# African Swine Fever Virus A528R Inhibits TLR8 Mediated NF-κB Activity by Targeting p65 Activation and Nuclear Translocation

**DOI:** 10.3390/v13102046

**Published:** 2021-10-11

**Authors:** Xueliang Liu, Da Ao, Sen Jiang, Nengwen Xia, Yulin Xu, Qi Shao, Jia Luo, Heng Wang, Wanglong Zheng, Nanhua Chen, François Meurens, Jianzhong Zhu

**Affiliations:** 1College Veterinary Medicine, Yangzhou University, Yangzhou 225009, China; xueliangaaa@foxmail.com (X.L.); da-ao@outlook.com (D.A.); Deepfisherr@outlook.com (S.J.); ydzjz1@yeah.net (N.X.); pangdunlee@gmail.com (Y.X.); sq970123@163.com (Q.S.); luojiaemail2021@yeah.net (J.L.); sdaulellow@163.com (H.W.); wanglongzheng@yzu.edu.cn (W.Z.); hnchen@yzu.edu.cn (N.C.); 2Comparative Medicine Research Institute, Yangzhou University, Yangzhou 225009, China; 3Joint International Research Laboratory of Agriculture and Agri-Product Safety, Yangzhou 225009, China; 4Jiangsu Co-Innovation Center for Prevention and Control of Important Animal Infectious Diseases and Zoonoses, Yangzhou 225009, China; 5BIOEPAR, INRAE, Oniris, 44307 Nantes, France; francois.meurens@inra.fr; 6Department of Veterinary Microbiology and Immunology, Western College of Veterinary Medicine, University of Saskatchewan, Saskatoon, SK S7N 5E2, Canada

**Keywords:** African swine fever virus, A528R protein, TLR8, NF-κB, p65

## Abstract

African swine fever (ASF) is mainly an acute hemorrhagic disease which is highly contagious and lethal to domestic pigs and wild boars. The global pig industry has suffered significant economic losses due to the lack of an effective vaccine and treatment. The African swine fever virus (ASFV) has a large genome of 170–190 kb, encoding more than 150 proteins. During infection, ASFV evades host innate immunity via multiple viral proteins. A528R is a very important member of the polygene family of ASFV, which was shown to inhibit IFN-β production by targeting NF-κB, but its mechanism is not clear. This study has shown that A528R can suppress the TLR8-NF-κB signaling pathway, including the inhibition of downstream promoter activity, NF-κB p65 phosphorylation and nuclear translocation, and the antiviral and antibacterial activity. Further, we found the cellular co-localization and interaction between A528R and p65, and ANK repeat domains of A528R and RHD of p65 are involved in their interaction and the inhibition of p65 activity. Therefore, we conclude that A528R inhibits TLR8-NF-κB signaling by targeting p65 activation and nuclear translocation.

## 1. Introduction

African swine fever (ASF) is mainly an acute febrile hemorrhagic disease that is highly contagious between domestic pigs and wild boars. In most cases, it is characterized by short onset and 100% acute mortality [1]. The clinical symptoms of ASF are similar to those of classical swine fever (CSF), both of which can only be distinguished by laboratory diagnosis. ASF lacks effective therapeutic drugs and vaccines and can only be prevented through biosafety measures. The high infectivity and mortality of domestic pigs due to ASF have caused great economic losses to the global pig industry [2].

African swine fever virus (ASFV), as the causative agent of ASF, belongs to a large linear DNA virus. Its genome length is about 170 to 193 kb and encodes 150 to 167 open reading frames [3]. The main cells infected by ASFV are monocytes and macrophages, megakaryocytes, and polymorphonuclear leukocytes in blood and bone marrow [4]. Studies have shown that after ASFV infects the host, it will regulate a variety of cellular pathways to escape the host’s innate immunity, mainly the production of type I interferons and inflammatory response via the actions of an array of viral proteins [5,6]. The multigene family of ASF (MGF) is located in the right 20 kb and the left 40 kb of the genome, which consists of MGFs 100, 110, 300, 360, and 505/530. MGFs affect virus virulence and/or replication, and the variation of MGFs results in the difference between strong and weak virulent strains of ASFV. Many studies have also shown that MGFs play an important role in modulating host innate immunity [7].

The innate immune system can detect the pathogen associated molecular patterns (PAMPs) of invading organisms through pattern recognition receptors (PRRs) so as to activate the host innate immunity. TLR8 belongs to the TLR7/8/9 family of Toll-like receptors, mainly distributed in myeloid monocytes, macrophages, etc. [8]. Activation of TLR8 recruits the homologous domain containing adaptor MyD88, which attracts and activates the protein kinases IRAK4 and IRAK2. These kinases activate the downstream binding molecule TRAF6, which activates the IKKα/β/γ complex leading to the activation of NF-κB and expression of inflammatory genes driven by NF-κB [9]. TLR8 is located in the endosome and the TLR8-NF-κB signaling pathway has antibacterial and antiviral effects [10]. We screened the ASFV China 2018/1 genomic ORFs with inhibitory effect on the TLR8 signaling pathway through TLR8-NF-κB stable reporter cell line [11]. Multiple proteins that inhibit TLR8 signaling have been identified, and A528R exhibited an obvious inhibitory effect on TLR8-NF-κB signaling.

A528R is also called MGF505-7R, one gene of the multi-gene family MGF505. It was originally found by sequencing in the restriction fragment *EcoR*I A-*Sal*I C of ASFV genome DNA. A528R-deleted ASFV had better IFN inducibility than wild-type ASFV and was fully attenuated in swine. The A528R gene, conserved across ASFV strains, has been identified as a common ASFV-specific T cell determinant and it has also been identified as a candidate to be included in future complex vaccine formulations [12]. It was initially identified as the inhibitor of poly I:C mediated IRF3 and NF-κB signaling, as well as type I IFN mediated signaling [13]. Recent reports showed that A528 inhibited cGAS-STING-IFN signaling as well as NLRP3 inflammasome signaling [14,15]. Although the inhibitory effect of A528R on NF-κB signaling has been reported for a long time, the mechanism by which A528R inhibits the NF-κB signaling pathway is not clear [16]. This study is to explore the inhibitory effect of A528R on TLR8-NF-κB signaling and its mechanism of action.

## 2. Materials and Methods

### 2.1. Cell Culture, Transfection, and Reagents

DMEM medium and RPMI 1640 medium were obtained from HyClone (HyClone Laboratories, Logan, UT, USA). Fetal bovine serum (FBS) was obtained from Gibco (Grand Island, NY, USA). Human embryonic kidney (HEK) 293T was maintained in DMEM supplemented with 10% fetal bovine serum, 1% penicillin–streptomycin solution (Gibco, Grand Island, NY, USA) at 37 °C in a humidified atmosphere of 5% CO_2_. Porcine alveolar macrophages (PAMs, 3D4/21) were cultured in RPMI 1640 medium, and the other conditions were the same. Transfection was performed by using the Lipofectamine 2000 (Invitrogen, Carlsbad, CA, USA) following the manufacturer’s instructions. The R848 was from MedChemExpress (Shanghai, China). The Mouse GFP mAb was from Absin (Shanghai, China). The Rabbit NF-κB p65 (AF5006) mAb and the Rabbit phospho-p65 (AF2006) mAb were from Affinity (Jiangsu, China). The Rabbit HA (3724S) mAb and the Rabbit GFP (2956) mAb were from Cell Signaling Technology (CST, Boston, MA, USA). The Mouse HA mAb and the Mouse GAPDH mAb were from Abclonal (WuHan, China). The Rabbit Histone H3 mAb were from Abcam (Cambridge, UK). HRP-anti mouse secondary antibody and HRP-anti rabbit secondary antibody were from Sangon Biotech (Shanghai, China). Goat Anti-Rabbit IgG Alexa Fluor 594 and Goat Anti-Mouse IgG Alexa Fluor 488 were from Thermo Fisher (Sunnyvale, CA, USA). VSV was stored and used regularly in the lab. *Staphylococcus aureus* was a gift from Dr. Heng Wang of Veterinary Hospital of Yangzhou University. Double-Luciferase Reporter Assay Kit were bought from TransGen Biotech (Beijing, China). MultiF Seamless Assembly Mix (RK21020) was bought from ABclonal (Wuhan, China). Protein A/G PLUS-Agarose was bought from Santa Cruz Biotechnology (sc-2003, CA, USA).

### 2.2. Gene Cloning and Mutation

The A528R gene was PCR amplified from pCMV-FLAG-A528R and then cloned into *EcoR*I and *EcoR*V sites of pCAGGS-HA vector using the MultiF Seamless Assembly Mix, with the recombinant plasmid named as pCAGGS-A528R-HA. For deletion of each five Ankyrin repeat (ANK) sequences of A528R, the mutation PCR primers were designed using QuikChange Primer Design website (https://www.agilent.com.cn, accessed on 16 September 2021) (Table 1). Five sequence confirmed deletion mutants are named A528R Δ54-83, Δ129-158, Δ261-290, Δ292-321, and Δ322-352, respectively. Porcine TLR8, MyD88, IKK-β, and p65 were cloned and preserved in our lab previously. Specifically, TLR8, MyD88, IKK-β, and p65 were PCR amplified and cloned into the *Bgl*II and *Kpn*I sites of pEGFP-N1 vector all by T4 ligation. NF-κB p65 N-terminal Rel homology domain (RHD, amino acids 19–306) and C-terminal non-Rel (amino acids 313–550) were both PCR amplified and cloned into the *Nhe*I and *Hind*III sites of pEGFP-N1 using seamless cloning. All the PCR primers used are shown in Table 1.

### 2.3. Dual-Luciferase Reporter (DLR) Promoter Assay

Transfection was performed using Lipofectamine 2000 when the cells grew to about 80% in 96-well plates. Cells in each well were co-transfected with reporter plasmids ELAM-firefly luciferase reporter (10 ng/well) (Fluc) and β-actin *Renilla* luciferase (Rluc) reporter (0.4 ng/well) together with the indicated plasmids, with the total DNA amount normalized to 50 ng by vector plasmids. Twenty-four hours post transfection, cells were stimulated with R848 (5 µg/mL) for another twelve hours. Cells were collected, lysed and luciferase activities were measured with Dual Luciferase Assay Kit (Vazyme, Nanjing, Jiangsu, China) in a Microplate Luminometer.

### 2.4. Nuclear and Cytoplasm Separation

The cytoplasmic and nuclear proteins were extracted using Nuclear and Cytoplasmic Protein Extraction Kit (Cat. no: P0028, Beyotime, Shanghai, China). The cells in the 12-well plates (3 × 10^5^ cells/well) were transfected with different plasmids for 24 h. After rinse with PBS, the cells were collected by pipetting and centrifugation. The collected cells were added with 200 µL of Cytoplasmic Protein Extraction Reagent, and violently vibrated followed by ice bath for 10 min. After high-speed centrifugation for 5 min, the supernatant was immediately collected with caution to obtain the cytoplasmic protein. The residual supernatant in the remained precipitation was completely removed and 50 µL nuclear protein extraction reagent was added to the precipitate. The nuclear suspension was violently vibrated followed by ice bath for 30 min. Then, the nuclear suspension was centrifuged, and the supernatant was collected to obtain soluble nuclear protein. The protein samples were mixed with 1× loading buffer and boiled for 10 min, then subjected for Western blotting.

### 2.5. Co-Immunoprecipitation (Co-IP) and Western Blotting (WB)

The cells in the 6-well plate (0.6–1.0 × 10^6^ cells/well) were transfected with different plasmids for 24 h. Cells were lysed in RIPA buffer (Cat. no: BL509A, Biosharp, Hefei, Anhui, China) for 15 min on ice. The lysate from centrifugation was pre-cleared with agarose beads by shaking for 4 h. After high-speed centrifugation, the pre-cleared lysate in supernatant was incubated with 1 µg target primary antibody by shaking at 4 °C overnight. Then 20 µL 50% agarose beads was added and incubated by shaking for further 4 h. The agarose beads were subsequently washed three times with RIPA buffer by centrifugation. Finally, the agarose beads were added 30–40 µL 2× loading buffer and boiled for 10 min to elute the bound proteins. The eluted proteins were separated by 8% or 12% SDS-PAGE and transferred to PVDF membranes. The membranes were sequentially blocked with 5% skim milk solution at room temperature (RT) for 2 h, probed with the indicated primary antibodies for 12 h at 4 °C, and then incubated with HRP-anti mouse or rabbit secondary antibodies for 1 h at RT. The protein signals were detected by ECL detection substrate (Biosharp, Hefei, Anhui, China) and imaging system (Tanon, Shanghai, China). In Co-IP experiment, the interaction between A528R and p65-GFP was subjected to overnight incubation shock using the Rabbit anti-HA mAb, followed by WB detection using the Mouse anti-GFP mAb and the Rabbit anti-HA mAb. Interaction of A528R and p65 mutants was subjected to overnight incubation shock using the Rabbit anti-HA mAb, followed by WB assay using the Mouse anti-GFP mAb and the Mouse anti-HA mAb. Interaction of p65 and A528R mutants was subjected to overnight shock incubation using the Mouse anti-HA mAb, followed by WB assay using the Rabbit anti-p65 mAb and the Rabbit anti-HA mAb.

### 2.6. Fluorescence Microscopy

PAMs grown on glass coverslip in 12-well plate (1.5 × 10^5^ cells/well) were transfected with A528R (1 µg) for 24 h, and then were stimulated with R848 (5 µg/mL) for 12 h. The cells were fixed with 4% paraformaldehyde at room temperature (RT) for 30 min, and permeabilized with 0.5% Triton X-100 for 20 min. After rinse with PBS, the cells were incubated with anti-p65 rabbit pAb (1:200), and next secondary antibodies Goat Anti-Rabbit IgG Alexa Fluor 594 (1:500). In another case, PAMs were co-transfected with A528R (1 µg) and p65-GFP (1 µg) for 24 h. These cells were sequentially stained with primary anti-HA rabbit pAb (1:200), and secondary antibodies Goat Anti-Rabbit IgG Alexa Fluor 594 (1:500). The stained cells were counterstained with 0.5 µg/mL 4′,6′-diamidino-2-phenylindole (DAPI, Beyotime, Shanghai, China) at 37 °C for 15 min to stain the cell nucleus. Lastly, PAMs were visualized under laser-scanning confocal microscope (LSCM, Leica SP8, Solms, Germany) at the excitation wavelength of 488 nm and 594 nm, respectively.

### 2.7. Bacteriostatic Experiment

PAMs were grown in 12-well plates (3 × 10^5^ cells/well) and transfected with plasmids after 24 h. The transfected cells were stimulated with R848 (5 µg/mL) for 12 h. Single *Staphylococcus aureus* colony on LB plate was picked and cultured overnight in 5 mL LB broth. The bacteria were collected by centrifugation when the culture OD600 = 1 (1OD600 approximately equals 1 × 10^9^ CFU/mL). The bacteria diluent was added to PAMs with a ratio of bacteria to cell 15:1 and for 1 h incubation. Next, the cell supernatant was discarded, and cells were washed with PBS 3 times, followed by incubation of RPMI 1640 containing penicillin (100 µg/mL), streptomycin (5 µg/mL), and 10% FBS for 6 h. Finally, the cells were collected by scraping and centrifugation and the harvested cells were lysed in 50 µL RIPA buffer. The cell lysates were diluted in PBS to 10^−5^, 10^−6^, and 10^−7^, respectively. The dilutions were spread on the LB agar plates and cultured overnight at 37 °C. At next day, the bacterial colony numbers were counted with a counter [17].

### 2.8. Statistical Analysis

Data were expressed as the mean ± SEM for each cellular experiment. The results were from one representative experiment of three similar experiments. Statistical analysis was performed by using Student’s *t*-test with GraphPad Prism v. 8.3. * *p* < 0.05; denote statistically significant and ns indicates no significance.

## 3. Results

### 3.1. A528R Interferes with the NF-κB Promoter Activity Mediated by TLR8

In order to explore the mechanism of ASFV evading host innate immunity, we used a porcine TLR8-NF-κB stable cell line to analyze the effects of ASFV proteins on the TLR8-NF-κB pathway [11]. From the screening, we identified several viral proteins with inhibitory effect on the TLR8-NF-κB pathway, among which A528R had the very significant effect. To further study the role of A528R in TLR8 mediated signal pathway, the effect of A528R on activations of TLR8 mediated NF-κB promoter was analyzed by using dual-luciferase reporter assay. ASFV A528R was co-transfected with the main signaling proteins along the TLR8-NF-κB pathway including porcine TLR8, MyD88, IKK-β, and p65 together with ELAM (NF-κB)-Luc reporter plasmid into 293T cells. Through ectopic expressions of the major proteins in the TLR8 signaling pathway to activate the downstream NF-κB, the influence of A528R on the signaling protein triggered NF-κB activation was measured by the reporter assay, so that the targets of A528R on the TLR8-NF-κB pathway could be determined. After 24 h of transfection, ELAM-Luc promoter activity was detected using dual-luciferase reporter assay. The results showed that R848 stimulated pTLR8 mediated NF-κB promoter activity was inhibited by A528R in a dose-dependent manner (Figure 1A). The ectopic expressions of porcine MyD88, IKKβ, and p65 could activate downstream NF-κB promoter, whereas in the presence of A528R, the NF-κB promoter activities by MyD88, IKKβ, and p65 were all decreased in dose-dependent manners (Figure 1B–D). The results indicated that p65 is very likely to be the target point of A528R for inhibition of TLR8 signaling, although it is also possible that other target points of the TLR8 signaling pathway are targeted by A528R. As a complementary experiment, we used R848 to stimulate endogenous TLR8 signaling in PAMs (3D4/21) and the transfected A528R could inhibit the NF-κB promoter activity mediated by TLR8 signal (Figure 1E).

### 3.2. A528R Inhibits the Phosphorylation of NF-κB p65 Downstream TLR8 Signaling

To validate the promoter assay results, we carried out the Western blotting experiment to detect the NF-κB p65 phosphorylation downstream TLR8 signaling. Similar to the NF-κB promoter assay, the R848 stimulated pTLR8 mediated p65 phosphorylation was suppressed by A528R in a dose-dependent manner (Figure 2A). The ectopic expressions of porcine MyD88, IKKβ and p65 activated the phosphorylations of p65, whereas, in the presence of A528R, the p65 phosphorylations were decreased in dose-dependent manners (Figure 2B–D). NF-κB nuclear entry is the main hallmark after TLR8 activation and the premise for inflammatory gene transcription. NF-κB p65, as a member of the classical NF-κB pathway, enters the nucleus after inhibitory κB (IκB) degradation caused by upstream activation signal. To detect the p65 nuclear translocation, the cells were fractionated and analyzed by Western blotting (Figure 2E). The results demonstrated that A528R did not affect the levels of p65 in the cytoplasm. However, A528R transfection significantly reduced the nuclear expressions of both phosphorylated p65 and p65 in response to R848 treatment (Figure 2E). These results prove that A528R inhibits the NF-κB p65 phosphorylation of the TLR8 pathway and also prevents p65 from entering the nucleus.

### 3.3. A528R Exhibits Cellular Co-Localization with p65 and Prevents p65 Nuclear Translocation

In order to study the effect of A528R on p65 nuclear translocation directly, the cellular localization of p65 in PAMs was observed by immunofluorescence microscopy. Without R848 stimulation, endogenous p65 was distributed in cytoplasm. Upon R848 stimulation of TLR8 signaling, the endogenous p65 protein was distributed in both the cytoplasm and nucleus (Figure 3A), whereas A528R transfection resulted in the aggregation of p65 in the cytoplasm and the almost disappearance of nuclear p65 (Figure 3A). These observations revealed the capacity of A528R to inhibit the nuclear localization of p65. Meanwhile, we transfected GFP tagged porcine p65 into PAMs and found that A528R and ectopic p65 were co-localized in cells (Figure 3B). Therefore, we reason that there may exist interaction between A528R and p65 which affects the nuclear entry of p65.

### 3.4. A528R Interacts with NF-κB p65 to Suppress its Promoter Activity

In order to further explore the relationship between A528R and p65, co-immunoprecipitation (Co-IP) were performed. The results show that there is an interaction between A528R and porcine p65 (Figure 4A). Based on the data retrieved from UniProt (https://www.uniprot.org/uploadlists/, accessed on 16 September 2021), porcine p65 is composed of N-terminal Rel homology domain (RHD) region and C-terminal non-RHD region. We cloned the p65 RHD region and non-RHD region into pEGFP-N1 vector, respectively. The Co-IP showed that there is an interaction between A528R and p65 RHD region and not p65 non-RHD (Figure 4B). According to the prediction of UniProt, there are five Ankyrin repeats (ANK) existing in A528, including amino acids 54–83, 129–158, 261–290, 292–321, and 322–352. We made the five individual deletion mutants of A528R, and the Co-IP results showed that all five deletion mutants had weakened interactions with p65 relative to full length A528R (Figure 4C). Accordingly, compared with the full length A528R, the five ANK deletion mutants all had lessened suppression to p65 activity in promoter assay (Figure 4D).

### 3.5. A528R Attenuated the Antiviral and Antibacterial Effects of TLR8 Signaling

NF-κB plays an important role in antibacterial and antiviral infection. In order to explore the effect of A528R on TLR8-NF-κB mediated anti-infection function, PAMs were transfected with A528R and one ANK deletion mutant, respectively, stimulated with R848, and then infected with either VSV or *S. aureus*. Upon R848 stimulation, VSV encoded GFP decreased substantially, observed by fluorescence microscopy (Figure 5A) and Western blotting (Figure 5B), suggesting the antiviral function by R848 stimulated TLR8-NF-κB signaling. In the presence of A528R, the viral GFP reversed back to a normal VSV infection level, whereas in the presence of A528R mutant, the viral GFP signal reversed but to a lesser degree (Figure 5A–B). On the other hand, the bacterial growth results showed that, compared with the bacteria in mock stimulated PAMs, the number of bacteria in R848 stimulated PAMs decreased obviously, suggesting the antibacterial function of TLR8-NF-κB signaling (Figure 5C–D). In the presence of A528R, the number of bacteria rebounded close to a normal infection level, whereas in the presence of A528R deletion mutant, the number of bacteria rebounded to a much lesser extent (Figure 5C–D). These results demonstrated that A528R can antagonize the anti-infection effect of TLR8-NF-κB signaling and the ANK domain of A528 is involved in such antagonism.

## 4. Discussion

ASFV causes acute hemorrhagic fever in domestic pigs and wild boars, and the mortality rate is very high. Due to the rapid development of global trade, African swine fever has spread in Eurasia, which has dealt a blow to the global pig industry [18]. Due to the lack of vaccine and effective treatment measures, ASF can only be prevented through biosafety [19]. During the long evolution, ASFV has developed various mechanisms to escape the innate immunity of the host. Understanding the immune escape mechanism of ASFV is of significance for the development of an effective vaccine and treatments [20].

In this study, we found that ASFV A528R/MGF505-7R suppresses TLR8-NF-κB signaling by targeting p65 activation and nuclear translocation. Until now, A528 has been reported to target not only IFN induction and signaling, but also inflammation occurrence. Specifically, in terms of IFN response, A528R promotes the expression of the autophagy-related protein ULK1 to degrade STING and suppress IFN induction [15]. A528R also interacts with IRF3 to inhibit its nuclear translocation and block type I IFN production [14]. In terms of inflammation, A528R interacts with IKKα to inhibit IL-1β gene transcription and with NLRP3 to block proIL-1β maturation and secretion [14]. In our study, we found A528R interacts with NF-κB p65 to inhibit NF-κB activation, but we did not exclude other signaling proteins upstream p65, such as IKKα/β/γ complex, which may be modulated by A528R. Therefore, our results are complementary with previous reports, and all these results suggest that A528R harbors a very broad-spectrum action on the innate immune response and prior attention should be drawn to it in the development of a gene-deletion attenuated ASF vaccine.

We show here how the Rel homology domain (RHD) of NF-κB p65 interacts with A528R. NF-κB refers to multiple dimers of Rel homology domain (RHD) containing proteins including Rel A (p65), c-Rel, Rel-B, p50, and p52, which are regulated by cellular signaling cascades that mediate the responses to inflammatory intercellular cytokines, pathogen exposure, and developmental signals [21]. The conserved RHD mediates NF-κB protein dimerization and DNA-binding for activation of gene transcription [22]. In quiescent cells, NF-κB is suppressed in cytoplasm by its inhibitor called inhibitory κB (IκB). IκB proteins belong to the Ankyrin repeat family due to the existence of a core series of structural repeats termed Ankyrin [23,24], which acts as the molecular architecture for protein recognition [25]. Interestingly, we found that A528R is an Ankyrin repeat protein and the five ANK domains are involved in the interaction with NF-κB p65. From this perspective, the logic is clear: ASFV utilizes A528R as a mimic of IκB to suppress NF-κB activation and inflammation.

Mammalian TLR8 is normally endosomal/lysosomal localized, and the physiological agonist is single stranded RNA (ssRNA) [26]. Then comes the question, how is TLR8 activated during infection by ASFV which is a DNA virus? ASFV enters the infected cells via clathrin mediated endocytosis (CME) pathway [27] and micropinocytosis pathway [28]. After entry into cells, the ASFV particles exist in cell early endosomes which traffic to lysosomes for maturation [29]. During the trafficking process, the viral particles are subjected to decapsidation and membrane fusion, leading to viral core release from lysosomes into cytosol [29]. The viral core will then be recruited into the perinuclear microtubule organizing center (MTOC), together with other components, to form a viral factory (VF) for viral replication and assembly [29]. Before DNA replication begins, some viral immediate early and early genes such as p30 gene (CP204L) are expressed [30]. The gene transcribed mRNA in endolysosomes will be recognized by the luminal TLR8. In addition, at late stage ASFV replication, the VF disrupts and reorganizes many cell organelles [31], and under this scenario TLR8 is likely to be activated by viral gene transcript mRNAs enriched in VF.

TLR8 signaling mainly induces the activation of NF-κB and plays an important role in antibacterial and antivirus infections [32]. We show here that TLR8 signaling triggered by agonist R848 exhibits anti-VSV and anti-*S. aureus* activity as expected. ASFV A528R interferes with the anti-VSV activity by R848-TLR8 signaling; not only that, but it also damages the anti-*S. aureus* activity by R848-TLR8. These results further confirm that A528R, as the immune evasion protein encoded by ASFV, promotes ASFV replication during infection. Additionally, A528R might promote co-infections of bacteria during ASFV infection. Even though we could not find any report of bacterial co-infection for ASFV currently, with the appearance of subacute and chronic ASF in the field [33,34], the bacteria co-infections might become apparent and will draw attention.

## 5. Conclusions

In conclusion, this study reveals that ASFV A528R interferes with the phosphorylation and nuclear translocation of NF-κB p65 through its interaction with p65, thereby inhibiting TLR8-NF-κB signaling pathway and its anti-infection function. The interaction between A528R and p65 is closely related to the ANK sites on A528R and directly linked to RHD of p65. The findings of this study complement our understanding of the immunosuppressive mechanism of A528R and provide new insights into the immune escape mechanism of ASFV.

## Figures and Tables

**Figure 1 viruses-13-02046-f001:**
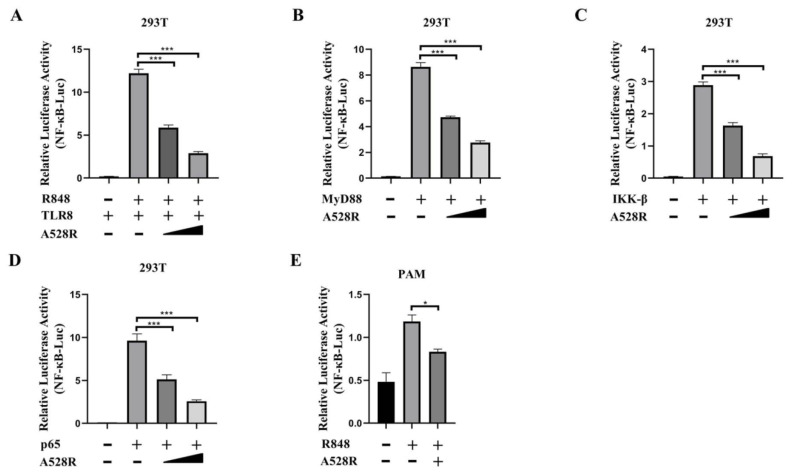
ASFV A528R weakens NF-κB promoter activity mediated by TLR8 signaling. (**A**) 293T cells grown in 96-well plates (3 × 10^4^ cells/well) were transfected by Lipofectamine 2000 with pTLR8 (30 ng), A528R (5 ng, 10 ng), plus reporters Fluc (10 ng) and Rluc (0.4 ng), which were normalized to 50 ng/well. At 24 h post transfection, cells were stimulated with R848 (5 µg/mL) for 12 h. (**B**–**D**) 293T cells grown in 96-well plate were transfected for 24 h with MyD88 (30 ng), A528R (5 ng, 10 ng) (**B**), IKK-β (30 ng), A528R (5 ng, 10 ng) (**C**), p65 (30 ng), A528R (5 ng, 10 ng) (**D**), plus reporters Fluc (10 ng) and Rluc (0.4 ng), which were normalized to 50 ng/well. (**E**) PAMs grown in 96-well plates (3 × 10^4^ cells/well) were transfected with A528R plasmids (20 ng), plus ELAM-Fluc (20 ng) and Rluc (0.4 ng) for 24 h, followed by R848 (5 µg/mL) stimulation for 12 h. The cells were collected for measurement of luciferase activities. Values represent the mean ± S.D; * and *** denote *p* < 0.05 and *p* < 0.001 versus the control group, respectively.

**Figure 2 viruses-13-02046-f002:**
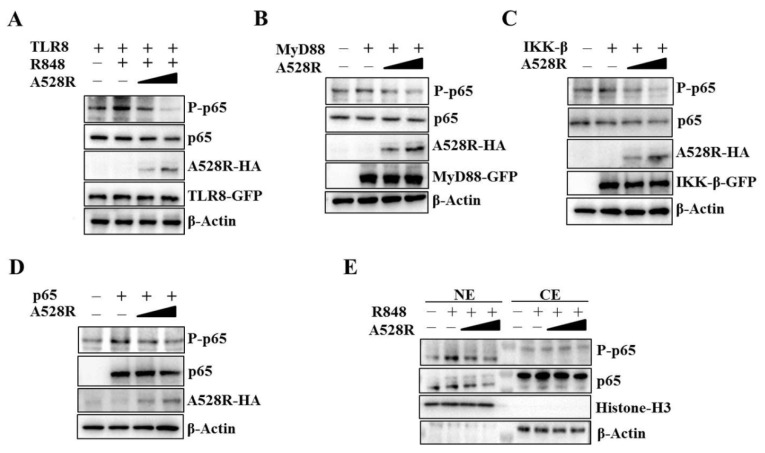
ASFV A528R inhibits the NF-κB p65 phosphorylation downstream TLR8 signaling. (**A**) 293T cells grown in 12-well plate (3 × 10^5^ cells/well) were transfected with A528R plasmid (0.25 µg, 0.5 µg), pTLR8 (0.5 µg) as indicated. Twenty-four hours post transfection, cells were stimulated with R848 (5 µg/mL) for 12 h, the expressions of p-p65, p65, A528R, and pTLR8 were analyzed by Western blotting. (**B**–**D**) 293T cells were transfected with A528R (0.25 µg, 0.5 µg) or vector (EV) control (0.5 µg), together with MyD88 (0.5 µg) (**B**), IKK-β (0.5 µg) (**C**), p65 (0.5 µg) (**D**). Twenty-four hours later, the expressions of p-p65, p65, A528R, MyD88, IKK-β, p65 were analyzed by Western blotting. (**E**) 293T cells grown in 12-well plate (3 × 10^5^ cells/well) were transfected with A528R (0.25 µg, 0.5 µg), pTLR8 (0.5 µg). Twenty-four hours post transfection, cells were stimulated with R848 (5 µg/mL) for 12 h. Cell lysates were fractionated into cytoplasmic and nuclear extracts, and the protein levels of p65 and p-p65 in different fractions were analyzed by Western blotting. The histone-H3 and GAPDH were also shown as the nuclear and cytoplasmic markers, respectively.

**Figure 3 viruses-13-02046-f003:**
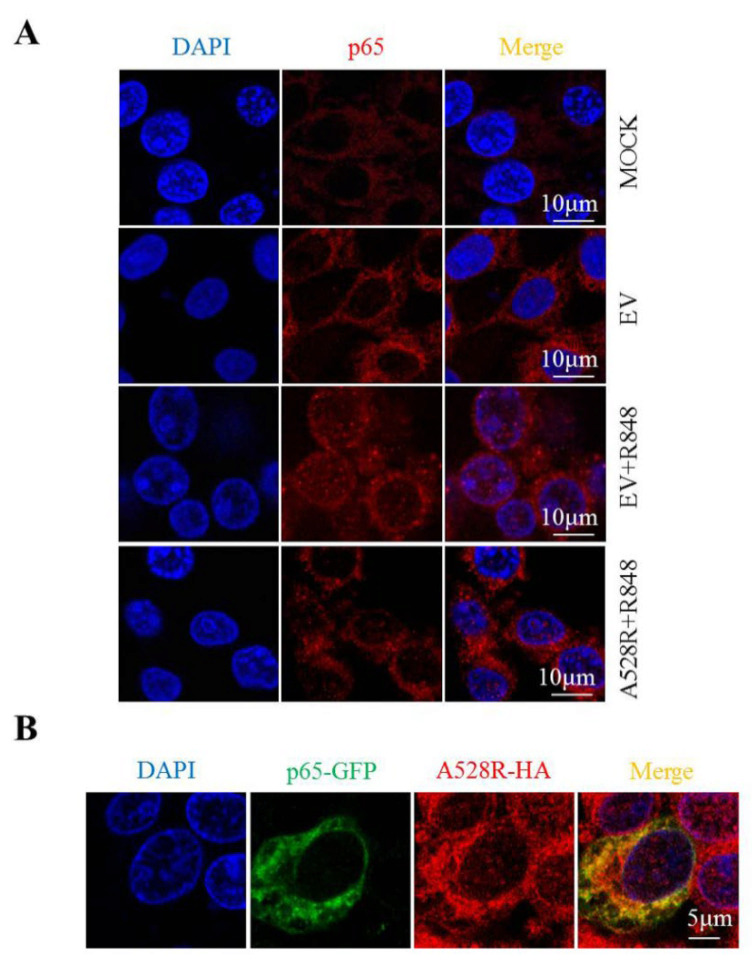
A528R co-localizes with p65 and inhibits nuclear translocation of p65. (**A**) PAMs grown on glass coverslip in 12-well plate (1.5 × 10^5^ cells/well) were transfected with A528R (1 µg) for 24 h, cells were stimulated with R848 (5 µg/mL) for another 12 h. Cells were fixed and stained followed by detection of p65 (red) and examined by confocal microscope. Magnification, 75×. (**B**) PAMs were transfected with p65-GFP and A528R-HA for 24 h. The cells were fixed and stained with anti-HA (red) antibody and examined by confocal microscope.

**Figure 4 viruses-13-02046-f004:**
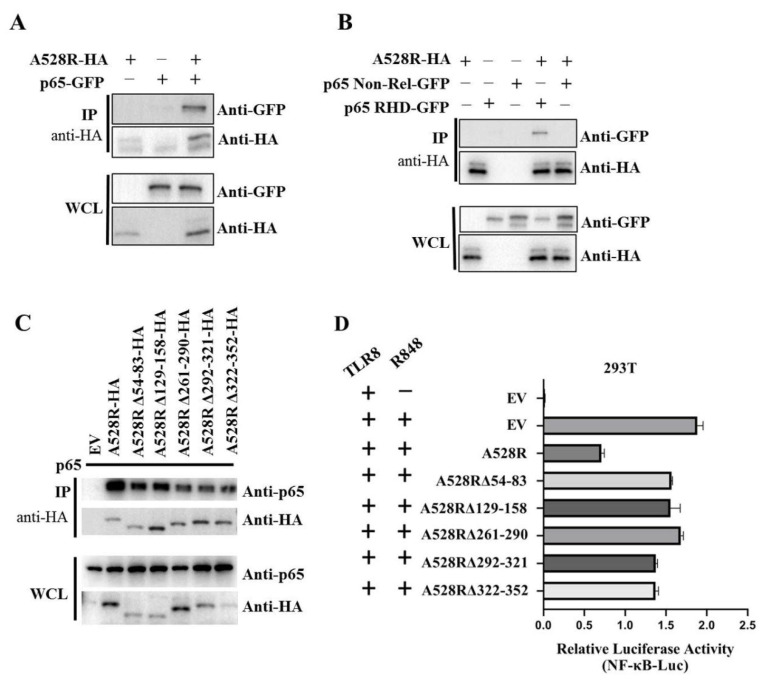
Interaction between A528R and p65 and analysis of the interaction sites. (**A**–**C**) 293T cells in 6-well plate (0.6–1 × 10^6^ cells/well) were transfected A528R-HA (0.5 µg) and p65-GFP (0.5 µg) (**A**), A528R-HA (0.5 µg) and p65 RHD (0.5 µg) or p65 non-Rel (0.5 µg) (**B**), A528R ANK deletion mutants (0.5 µg each) (**C**) for 24 h. The cell lysates were subjected for immunoprecipitation and subsequent Western blotting using the indicated antibodies. (**D**) 293T cells grown in 96-well plate (3 × 10^4^ cells/well) were transfected by Lipofectamine 2000 with pTLR8 (30 ng), A528R (10 ng), or each A528R ANK deletion mutants (10 ng) plus reporters Fluc (10 ng) and Rluc (0.4 ng), which were normalized to 50 ng/well. At 24 h post transfection, cells were stimulated with R848 (5 µg/mL) for 12 h, followed by the measurement of luciferase activities.

**Figure 5 viruses-13-02046-f005:**
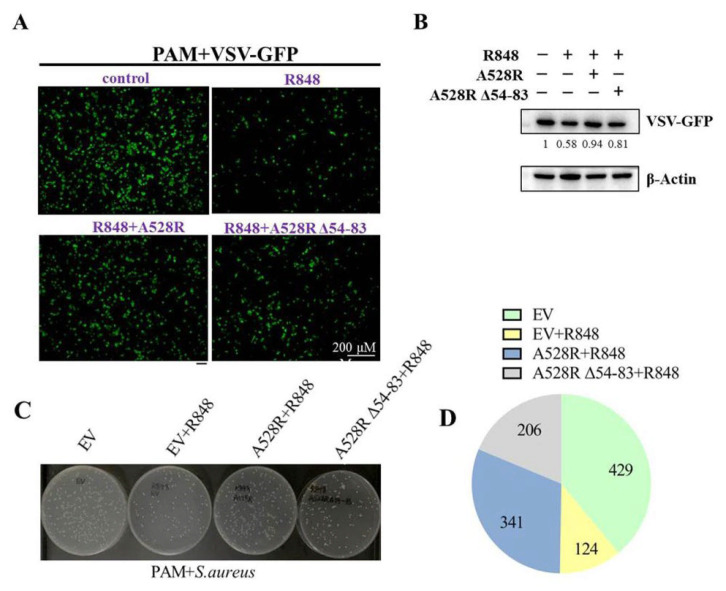
A528R inhibits the antiviral and antibacterial effects of TLR8-NF-κB signaling. (**A**–**B**) PAMs grown on glass coverslips in 12-well plate (3 × 10^5^ cells/well) were transfected with A528R-HA (1 µg), A528RΔ54-83 (1 µg), or empty vector. At 24 h post transfection, cells were stimulated with R848 (5 µg/mL) for 12 h. The cells were infected with VSV-GFP at the MOI 0.01 for 12 h. The VSV replicative GFP signals were observed by fluorescence microscopy (**A**) and the GFP expression was analyzed by Western blotting. The GFP gray density were quantified, and the values were shown under the respective GFP bands after normalization of β-action (**B**). (**C**–**D**) PAMs grown in 12-well plate (3 × 10^5^ cells/well) were transfected and treated with R848, as in A–B. The cells were infected with *Staphylococcus aureus* at MOI of 15. At 6 h post infection, the cells were harvested and lysed. The diluted lysates were spread on LB agar plate. The representative picture of bacterial colonies on overnight grown plates were shown (**C**), and the colony numbers of each sample were counted and shown as the chart form (**D**).

**Table 1 viruses-13-02046-t001:** PCR primers used in this study.

Primer Name	Primer Sequences (5′-3′)
A528R-F	TGTCTCATCATTTTGGCAAA*GAATTC*ATGTTCTCCCTTCAGG
A528R-R	ATCGTATGGGTAGCTGGT*GATATC*ATACATGGCATACTCCAA
A528RΔ54-83-F	CTTATAGAGCATGATCTTACTCTTGCCATCATAGGAGCTTTGAG
A528RΔ54-83-R	CTCAAAGCTCCTATGATGGCAAGAGTAAGATCATGCTCTATAAG
A528RΔ129-158-F	CGAAAAATGTCATGATTTAAGCCTTCTATTTAGGCAACAAATTCAAGGAC
A528RΔ129-158-R	GTCCTTGAATTTGTTGCCTAAATAGAAGGCTTAAATCATGACATTTTTCG
A528RΔ261-290-F	GGAAATTTTAAATTATGGTGGGAATATTGAAAGAATGTTGCATCTGGCT
A528RΔ261-290-R	AGCCAGATGCAACATTCTTTCAATATTCCCACCATAATTTAAAATTTCC
A528RΔ292-321-F	CAAAAGAATATACCCCATAAAACCATTGTTAAAAAGTTGTTAGAACATGTAGTG
A528RΔ292-321-R	CACTACATGTTCTAACAACTTTTTAACAATGGTTTTATGGGGTATATTCTTTTG
A528RΔ322-352-F	GAACTTGTTACTATCTTACATAAATTACAAGGTGAAAAATTTGACAAGATATGTCAAAGAT
A528RΔ322-352-R	ATCTTTGACATATCTTGTCAAATTTTTCACCTTGTAATTTATGTAAGATAGTAACAAGTTC
pTLR8-F	GA*AGATCT*ATGACCCTTCACTTTTTGCTCCTGACC
pTLR8-R	CGG*GGTACC*TTACTTAATGGAATTGACATACAAAC
pMyD88-F	AGCGCTACCGGACTC*AGATCT*ATGGCTGCAGGAGGCTCCG
pMyD88-R	ATCCCGGGCCCGC*GGTACC*GTGGGCAGGGATAGGGCCCTG
pIKK-β-F	AGCGCTACCGGACTC*AGATCT*ATGAGCTGGTCACCTTCCCTGA
pIKK-β-R	ATCCCGGGCCCGC*GGTACC*GTCGAGGCCTGCTCCAGGCT
pP65-F	AGCGCTACCGGACTC*AGATCT*ATGGACGACCTCTTCCCCCTCA
pP65-R	ATCCCGGGCCCGC*GGTACC*GTGGAGCTGATCTGACTCAGAAGGGC
p65 RHD-F	TGAACCGTCAGATCC*GCTAGC*CCCTATGTGGAGATCATCGAG
p65 RHD-R	CAGAATTCG*AAGCTT*GAGCTCATAGGTCCTTTTGCGTTTCTC
p65 Non-Rel-F	TGAACCGTCAGATCC*GCTAGC*GAGACCTTTAAGAGCATCATG
p65 Non-Rel-R	CAGAATTCG*AAGCTT*GAGCTCGGAGCTGATCTGACTCAGAAG

Note: the restriction sites in the primer sequences are italic and underlined.

## Data Availability

The data presented in this study are available in insert article.

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
