# Peer review of "African Swine Fever Virus A528R Inhibits TLR8 Mediated NF-κB Activity by Targeting p65 Activation and Nuclear Translocation"

_viruses, 2021, doi:10.3390/v13102046_

Round 1
Reviewer 1 Report
Since its arrival in Georgia in 2007, African swine fever has spread rapidly, through Russia and Europe, and more recently to America. Although the virus is harmless to humans, the current outbreak of African swine fever could have serious global repercussions for food security and economic stability. China – home to half the world’s pig population – has lost a third of its pigs to the disease, possibly more. The disease is carried by pigs, wild boar and soft-bodied ticks, and there is currently no vaccine or cure. However, according to current knowledge , containment is possible through a combination of biosecurity, government and farmer action along with information and awareness from everyone. That's why ASFV-related studies are highly desirable and important.
The authors of this important and well-written paper have established that A528R can sup-21 press TLR8-NF-κB signaling pathway, including the inhibition of downstream promoter activity, 22 NF-κB p65 phosphorylation and nuclear translocation, and the antiviral and anti-bacterial activity. Additionally, it has been shown that the cellular co-localization and interaction between A528R and p65, and ANK 24 repeat domains of A528R and RHD of p65 are involved in their interaction and the inhibition of p65 25 activity. The results of this study complement the understanding of the immunosuppressive mechanism of A528R and provide new insights into the immune es-385 cape mechanism of ASFV.
Contents of all sections are appropriate and adequate. Materials and methods used in the study are adequately and minutely described . Results are well presented in the manuscript as well as discussion which is comprehensive. Conclusions were justified by the obtained results and correspond to the aim of the study.
Author Response
Thanks for your appreciation of our work. Our manuscript has been updated.
Reviewer 2 Report
Liu et la. described their works on studying the role of ASFV-A528R gene function and their results indicated that A528R inhibits TLR8-NF-κB signaling by targeting p65 activation and nuclear translocation. This paper was well written, however, there are some points needed further clarification.
Comments:
- The basic information and characteristis of African Swine Fever Virus A528R should offer more information and let the readers understand the importance of A528R and the advantage or disadvantage when the virus carrying this gene. Please try to add some additional information in the abstract and introduction.
- Lines 195-197, the purpose to use the dual-luciferase assay did not explain well. Please clarify this part.
- Lines 203-207, the current offered evidences still could not specifically point that p65 is the main target of A528R and which might affect the following results that A528R regulated p65 or more upper signaling factor.
- In Fig.3B, the fluorescent image of co-localization of p65 and A528R was not matched, some major particles did not co-localize well to show the yellow color.
- In Fig.4C, the authors indicated that five individual deletion mutants of A528R were made and the Co-IP results showed that all five 276 deletion mutants had weakened interactions with p65 relative to full length A528R. The IP data showed that expression of the truncated A528R was not so equal and their effects on p65 were also not the same. Please try to discuss this difference. In addition, the lane 1 on this figure is EV transfection ?
- In Fig.5, it is not clear why the authors chose VSV or S. aureus as the candidates for testing A528R mediated antiviral and anti-bacterial effects?
Author Response
Comments:
The basic information and characteristic of African Swine Fever Virus A528R should offer more information and let the readers understand the importance of A528R and the advantage or disadvantage when the virus carrying this gene. Please try to add some additional information in the abstract and introduction.
Answer: Thanks for your suggestion, we added the A528R background information in both Abstract and Results.
Lines 195-197, the purpose to use the dual-luciferase assay did not explain well. Please clarify this part.
Answer: We added the following description to clarify the purpose of the dual-luciferase assay. “Through ectopic expressions of the major proteins in the TLR8 signaling pathway to activate the downstream NF-kB, the influence of A528R on the signaling protein triggered NF-kB activation was measured by the reporter assay, so that the targets of A528R on TLR8-NF-κB pathway could be determined.”
Lines 203-207, the current offered evidences still could not specifically point that p65 is the main target of A528R and which might affect the following results that A528R regulated p65 or more upper signaling factor.
Answer: At present, our work cannot exclude the possibility that A528R has multiple targets in the TLR8 signaling pathway. Searching of other targets of A528R will be the focus of our future work. The focus of our current work was to reveal the interaction between A528R and p65, and the inhibitory effect of A528R on p65 activation and nuclear translocation. Further, the ANK domains of A528R and RHD domain of p65 are involved in their interaction, and the interaction is important for the inhibitory effects of A528R on p65 activation and TLR8 signaling mediated anti-infections.
In Fig.3B, the fluorescent image of co-localization of p65 and A528R was not matched, some major particles did not co-localize well to show the yellow color.
Answer: We have replaced with a new co-localization images of A528R and p65, to show the co-location more clearly and intuitively.
In Fig.4C, the authors indicated that five individual deletion mutants of A528R were made and the Co-IP results showed that all five 276 deletion mutants had weakened interactions with p65 relative to full length A528R. The IP data showed that expression of the truncated A528R was not so equal and their effects on p65 were also not the same. Please try to discuss this difference. In addition, the lane 1 on this figure is EV transfection ?
Answer: The expression amount of A528R deletion mutants in the experiment is different to a certain degree. After enriching A528R and its deletion mutants using IP with anti-HA antibody, we could see that the amount of five A528R deletion mutants in IP sample are the same as or a little bit higher than that of wild type A528R. In the same IP samples, we observed the expression of p65, and the amount of Co-IP p65 by various A528R mutants are less than that by wild type A528R, reflecting the attenuated interaction between A528R deletion mutants and p65. In general, the experimental results are in line with our experimental expectations.
In addition, the first lane in Figure 4C is empty vector (EV) transfection control. Due to our carelessness, we didn’t include the EV lane of the anti-HA blot in the IP samples; we are very sorry for that, and now it is updated and corrected.
In Fig.5, it is not clear why the authors chose VSV or S. aureus as the candidates for testing A528R mediated antiviral and anti-bacterial effects?
Answer: The reason why we used VSV and Staphylococcus aureus for experiments is to activate the cellular TLR8-NF-κB signaling pathway as much as possible. On this basis, the inhibitory effect of A528R would be relevant and convincing. Specifically, VSV belongs to single stranded (ss) RNA virus, and can be recognized by the TLR8 receptor, activating NF-κB signal pathway. In terms of Staphylococcus aureus, its RNA is mainly sensed by TLR8, which has been reported by several reports (J Immunol. 2015 Aug 1;195(3):1100-11; EMBO Rep. 2015 Dec;16(12):1656-63; Front Immunol. 2019 May 31;10:1209). For the above considerations, we finally chose VSV or S. aureus as the candidates for testing the effects of A528R on TLR8 signaling mediated antiviral and anti-bacterial function.
Reviewer 3 Report
The manuscript by Liu et al. describes the ASFV A528R/MGF505-7R suppresses TLR8-NF-kB signaling by targeting p65 activation adn nuclear translocation.
The paper is well written, and the technical elements are explained in a way to allow the reader to easily understand the logics. I have only minor suggestions to improve the quality of the manuscript:
Ref 13 and Ref 20 are same, need to be corrected.
Author Response
Thanks for your appreciation, the reference has been corrected. The manuscript has also been updated.